# Occurrence of High Levels of Cefiderocol Resistance in Carbapenem-Resistant *Escherichia coli* before Its Approval in China: a Report from China CRE-Network

Qi Wang,[a] Longyang Jin,[a] Shijun Sun,[a] Yuyao Yin,[a] Ruobing Wang,[a] Fengning Chen,[a] Xiaojuan Wang,[a] Yawei Zhang,[a] Jun Hou,[b] Yumei Zhang,[c] Zhijie Zhang,[d] Liuchun Luo,[e] Zhusheng Guo,[f] Zhenpeng Li,[g] Xin Lin,[h] Lei Bi,[i] ⊙Hui Wang[a]

[a]Department of Clinical Laboratory, Peking University People's Hospital, Beijing, China

[b]Department of Clinical Laboratory, The Third Hospital of Mianyang, Sichuan, China

[c]Department of Clinical Laboratory, People's Hospital of Zunhua, Hebei, China

[d]Department of Clinical Laboratory, Shengjing Hospital of China Medical University, Shenyang, China

[e]Department of Clinical Laboratory, Liuzhou People's Hospital, Guangxi, China

[f]Department of Clinical Laboratory, Dongguan Donghua Hospital, Guangdong, China

[g]Department of Clinical Laboratory, Wanbei Coal-Electricity Group General Hospital, Anhui, China

[h]Department of Clinical Laboratory, Nanjing First Hospital, Nanjing Medical University, Nanjing, China

[i]Department of Clinical Laboratory, Zibo Central Hospital, Shandong, China

Qi Wang, Longyang Jin, and Shijun Sun contributed equally to this article. Author order was determined by in order of decreasing seniority.

**ABSTRACT** Cefiderocol has been approved in the United States and Europe but not in China. We aim to evaluate carbapenem-resistant *Enterobacterales* (CRE) susceptibility to cefiderocol to provide baseline data and investigate the resistance mechanism. From 2018 to 2019, 1,158 CRE isolates were collected from 23 provinces and municipalities across China. The MICs of antimicrobials were determined via the agar dilution and broth microdilution methods. Whole-genome sequencing was performed for 26 cefiderocol-resistant *Escherichia coli* isolates to investigate the resistance mechanism. Clone transformations were used to explore the function of *cirA*, *pbp3*, and $bla_{NDM-5}$ in resistance. Among the 21 antimicrobials tested, aztreonam-avibactam had the highest antibacterial activity (98.3%), followed by cefiderocol (97.3%) and colistin (95.3%). A total of 26 *E. coli* isolates harboring New Delhi metallo-beta-lactamase 5 (NDM-5) showed high levels of cefiderocol resistance, of which sequence type 167 (ST167) accounted for 76.9% (20/26). We found 4 amino-acid insertions (YRIN/YRIK) at position 333 of penicillin-binding protein 3 (PBP3) in the 26 *E. coli* isolates, and 22 isolates had a siderophore receptor *cirA* premature stop codon. After obtaining the wild-type *cirA* supplementation, the MIC of the transformants decreased by 8 to 16 times in two cefiderocol-resistant isolates. A cefiderocol-susceptible isolate harboring NDM-5 has an MIC increased from 1 $\mu$g/mL to 64 $\mu$g/mL after *cirA* deletion, and the MIC decreased from 64 $\mu$g/mL to 0.5 $\mu$g/mL after $bla_{NDM-5}$ deletion. The MIC of the *E. coli* DH5$\alpha$, from which the *pbp3* mutant was obtained, increased from 0.064 $\mu$g/mL to 0.25 $\mu$g/mL. Cefiderocol showed activity against most CRE in China. The resistance of ST167 *E. coli* to cefiderocol is a combination of the premature stop codon of *cirA*, *pbp3* mutation, and $bla_{NDM-5}$ existence.

**IMPORTANCE** Cefiderocol, a new siderophore cephalosporin, has been approved in the United States and Europe but not in China. At present, there are almost no antimicrobial susceptibility evaluation data on cefiderocol in China. We evaluated the *in vitro* susceptibility of 1,158 strains of carbapenem-resistant *Enterobacterales* to cefiderocol and other antibiotics. We found that a high proportion of *Escherichia coli* showed high-level resistance to cefiderocol. Whole-genome sequencing (WGS) and molecular cloning experiments confirmed that the synergistic effect of the *cirA* gene premature stop codon, $bla_{NDM-5}$ existence, and the *pbp3* mutation is associated with high levels of cefiderocol resistance.

Address correspondence to Hui Wang, whuibj@163.com.

The authors declare no conflict of interest.

**KEYWORDS** cefiderocol, carbapenem-resistant *Enterobacterales*, *cirA*, *bla*$_{NDM-5}$, *pbp3*

Enterobacterales are important pathogens in hospital- and community-acquired infections and can cause serious infectious diseases, including bacteremia, pneumonia, and liver abscesses (1). Due to the widespread use of antimicrobial agents in clinical treatment, the occurrence of antimicrobial resistance and the number of carbapenem-resistant *Enterobacterales* (CRE) have increased (2, 3). CRE represent the most urgent threat category of antimicrobial resistance in the latest 2019 U.S. Centers for Disease Control and Prevention antibiotic resistance report (4). Only a few antimicrobial agents are used for treating CRE infections in China, including tigecycline, colistin, and ceftazidime-avibactam. However, these drugs are not always active. Tigecycline is not useful for all types of infection, especially not for bloodstream infections. However, intraabdominal infections, especially those affecting the bile duct system, can be treated successfully, as the concentration in bile is comparably high. The nephrotoxicity of colistin is of considerable concern in the treatment process. Colistin also has neurotoxicity, which is clinically relevant (5). In China, ceftazidime-avibactam was approved by the National Medical Products Administration in 2019; however, it is ineffective against metallo-beta-lactamases bearing Gram-negative bacilli. Therefore, CRE infections represent a significant challenge to health care in China, and the development and application of new antimicrobial agents are urgently required.

Cefiderocol, a novel catecholamine-siderophore cephalosporin, was approved by the U.S. Food and Drug Administration (FDA) and the European Medicines Agency in 2019 and 2020, respectively. In the 1980s, Miller's team first proposed a "Trojan horse" molecule-based antimicrobial treatment strategy (6), which couples an antimicrobial agent with the functional group of a siderophore and uses the siderophore transport system to deliver the active ingredients of the antimicrobial agent to the antimicrobial-binding site (7). Cefiderocol binds to ferric ions ($Fe^{3+}$) and is transported to the periplasmic space by TonB-dependent transporters (TBDTs) on the outer membrane of the bacteria, where it acts on penicillin-binding proteins (PBPs), subsequently inhibiting cell wall synthesis (8). This unique Trojan horse strategy also overcomes the permeability-related drug resistance mechanism associated with the loss of bacterial outer membrane pore protein and the overexpression of drug resistance-related efflux pumps (9).

Data from a continuous international antimicrobial susceptibility-monitoring project on cefiderocol, SIDERO-WT, shows that during a period of monitoring from 2014 to 2016, the susceptibility rate of most Gram-negative bacilli to cefiderocol exceeded 95% (10–12). In February 2021, a phase III clinical study using cefiderocol to treat severe infections caused by carbapenem-resistant Gram-negative bacteria showed that the clinical efficacy and microbiological clearance effect of cefiderocol are not inferior to those of the best treatment options currently available for clinical selection (13). As the first siderophore cephalosporin to pass a phase III clinical trial, cefiderocol shows potential benefit in CRE treatment. Therefore, this study aims to conduct antimicrobial susceptibility testing (AST) of cefiderocol and other antimicrobials for CRE isolates collected from multiple centers in China. We evaluated the *in vitro* antimicrobial susceptibility of CRE isolates to cefiderocol and other drugs and revealed the main resistance mechanism of China's current cefiderocol-resistant isolates.

## RESULTS

**Distribution of CRE isolates and types of carbapenemase produced.** Among the CRE isolates collected from 2018 to 2019 across China, *Klebsiella pneumoniae* (798/1,158, 68.9%) accounted for the highest proportion, followed by *E. coli* (181/1,158, 15.6%), *Enterobacter cloacae* complex (108/1,158, 9.3%), *Klebsiella oxytoca* (23/1,158, 2%), *Klebsiella aerogenes* (20/1,158, 1.7%), *Serratia marcescens* (14/1,158, 1.2%), *Citrobacter freundii* (12/1,158, 1%), *Citrobacter braakii* (1/1,158, 0.1%), and *Morganella morganii* (1/1,158, 0.1%).

The results of the modified carbapenem inactivation method (mCIM) test showed that 1,003 strains out of 1,158 strains of CRE were positive, accounting for 86.6%. PCR and sequencing analysis showed that 990 out of 1,158 CRE isolates had a positive

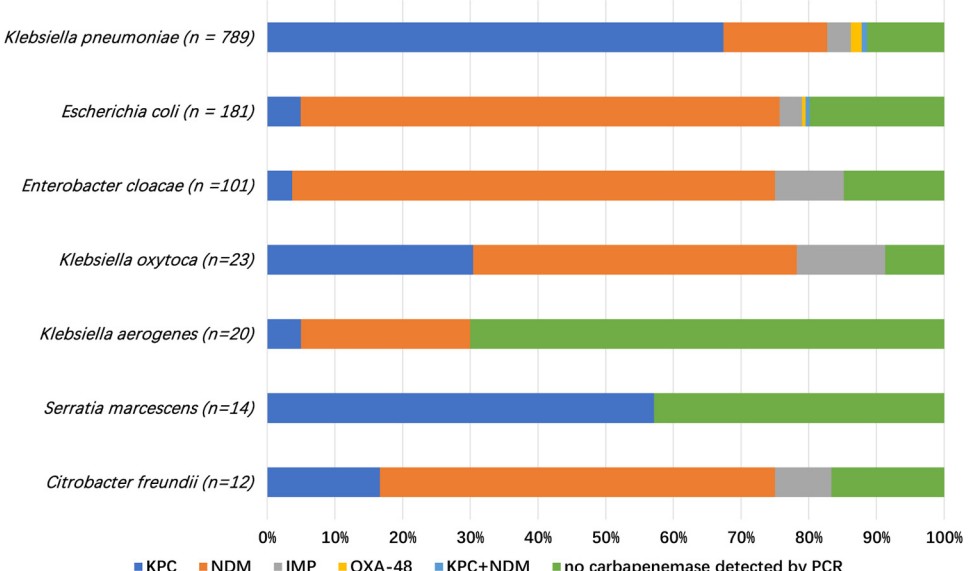

**FIG 1** Distribution of the most frequent carbapenemase carried by all CREs in this study.

result in the PCR for carbapenemases. Among the *K. pneumoniae* isolates, *K. pneumoniae* carbapenemase (KPC) was the main carbapenemase produced, accounting for 67.4% (538/798), followed by New Delhi metallo-beta-lactamase (NDM), accounting for 15.3% (122/798), and imipenemase (IMP), which accounted for 3.5% (28/798) of the isolates (Fig. 1). Six isolates (0.8%) of *K. pneumoniae* carried KPC and NDM genes. Among the carbapenem-resistant *E. coli*, 70.7% (128/181) of the isolates harbored NDM, 5.0% (9/181) harbored KPC, and 3.3% (6/181) harbored IMP. Among the carbapenem-resistant *E. cloacae* complex, 71.3% (77/108) of the isolates harbored NDM, 10.2% (11/108) harbored IMP, and 3.7% (4/108) harbored KPC.

**AST results.** The AST results of the major species and carbapenemases gene are shown in Tables 1 and 2. Among the 21 antimicrobial agents tested against 1,158 CRE isolates, the isolates showed the highest susceptibility to aztreonam-avibactam (98.3%), followed by cefiderocol (97.3%), colistin (95.3%), tigecycline (91.7%), ceftazidime-avibactam (66.7%), and amikacin (58.9%).

Carbapenem-resistant *K. pneumoniae* isolates showed highest susceptibility to cefiderocol (99.7%), followed by aztreonam-avibactam (98.5%), colistin (96.9%), tigecycline (90.2), and ceftazidime-avibactam (81.3%). Carbapenem-resistant *E. coli* showed highest susceptibility to colistin (99.4%), followed by tigecycline (98.3%), aztreonam-avibactam (95.6%), cefiderocol (85.1%), and fosfomycin (80.1%).

KPC-producing CRE isolates showed the highest susceptibility to cefiderocol (99.6%), aztreonam-avibactam (99.5%), ceftazidime-avibactam (97.4%), colistin (96%), and tigecycline (92.3%). NDM-producing CRE isolates showed the highest susceptibility to colistin (97.7%), aztreonam-avibactam (97.2%), tigecycline (93.4%), cefiderocol (92%), fosfomycin (83.5%), and amikacin (79.8%).

**Characteristics of cefiderocol-resistant *E. coli*.** Among the 1,158 CRE isolates, 30 isolates were resistant to cefiderocol, comprising 26 *E. coli*, 2 *K. pneumoniae*, and 2 *E. cloacae* complex isolates. We evaluated the phenotypic characteristics and resistance mechanisms of the *E. coli* isolates which represented the majority. Table 3 shows the general characteristics of the 26 cefiderocol-resistant *E. coli* isolates. Chronologically, 13 of the 26 cefiderocol-resistant isolates were isolated in 2018 and 13 in 2019. Geographically, 26 isolates were collected from 9 provinces in China; 9 isolates were from Sichuan, 5 were from Shandong, 3 were from Guangxi, 3 were from Hebei, and 2 were from Beijing, one each from Jiangsu, Shanxi, Hunan, and Guangdong, respectively. A total of 6 different multilocus sequence types (MLSTs) were identified, among

**TABLE 1** Antimicrobial susceptibility testing results of all CRE isolates from 2018 to 2019 in the China CRE-Network[a]

| Antibiotic | All isolates ($n = 1{,}158$) | | | | Klebsiella pneumoniae isolates ($n = 789$) | | | | Escherichia coli isolates ($n = 181$) | | | | Enterobacter cloacae isolates ($n = 101$) | | | |
|---|---|---|---|---|---|---|---|---|---|---|---|---|---|---|---|---|
| | %R | %S | $MIC_{50}$ | $MIC_{90}$ | %R | %S | $MIC_{50}$ | $MIC_{90}$ | %R | %S | $MIC_{50}$ | $MIC_{90}$ | %R | %S | $MIC_{50}$ | $MIC_{90}$ |
| Amikacin | 39.5 | 58.9 | 8 | 256 | 49 | 49.4 | 32 | 256 | 27.1 | 72.4 | 8 | 256 | 10.2 | 87 | 4 | 256 |
| Aztreonam | 82.6 | 14.5 | >256 | >256 | 91.2 | 7.1 | >256 | >256 | 66.3 | 27.6 | 64 | >256 | 63 | 31.5 | 32 | 256 |
| Aztreonam/avibactam | 0.9 | 98.3 | 0.25 | 1 | 1 | 98.5 | 0.5 | 1 | 1.1 | 95.6 | 0.125 | 2 | 0 | 100 | 0.125 | 0.5 |
| Cefepime | 89.1 | 6.2 | 128 | >256 | 92.7 | 4.4 | 128 | 256 | 85.6 | 7.7 | 256 | >256 | 87 | 8.3 | 64 | 256 |
| Cefiderocol | 2.7 | 97.3 | 2 | 2 | 0.3 | 99.7 | 1 | 2 | 14.4 | 85.1 | 2 | 64 | 1.9 | 98.1 | 2 | 2 |
| Cefoperazone/sulbactam | 89.7 | 6.5 | >256 | >256 | 93.1 | 4.6 | >256 | >256 | 86.2 | 8.3 | >256 | >256 | 86.1 | 9.3 | >256 | >256 |
| Cefotaxime | 95.6 | 4.2 | 256 | >256 | 96.4 | 3.4 | 256 | >256 | 96.1 | 3.9 | >256 | >256 | 96.3 | 3.7 | >256 | >256 |
| Cefoxitin | 91.9 | 5.5 | 256 | >256 | 92.6 | 4.8 | 256 | >256 | 87.3 | 11.6 | >256 | >256 | 98.1 | 1.9 | >256 | >256 |
| Ceftazidime | 92.2 | 6.4 | >256 | >256 | 94.4 | 4.3 | >256 | >256 | 91.2 | 6.6 | >256 | >256 | 92.6 | 7.4 | >256 | >256 |
| Ceftazidime/avibactam | 33.3 | 66.7 | 2 | >256 | 18.7 | 81.3 | 2 | >256 | 68 | 32 | >256 | >256 | 80.6 | 19.4 | >256 | >256 |
| Chloramphenicol | 55.5 | 33.2 | 32 | >256 | 62.2 | 26.9 | 32 | >256 | 43.6 | 43.1 | 16 | >256 | 39.8 | 51.9 | 8 | >256 |
| Ciprofloxacin | 81.9 | 13.6 | 64 | 128 | 84.8 | 11.9 | 64 | 128 | 86.7 | 7.7 | 64 | 128 | 63.9 | 21.3 | 2 | 128 |
| Colistin | 4.1 | 95.9 | 0.125 | 0.5 | 3.4 | 96.6 | 0.25 | 0.5 | 0.6 | 99.4 | 0.25 | 0.5 | 8.3 | 91.7 | 0.125 | 0.5 |
| Ertapenem | 89.6 | 8.2 | 64 | >256 | 93.5 | 5.4 | 128 | >256 | 82.9 | 13.8 | 32 | 128 | 87 | 7.4 | 8 | 64 |
| Fosfomycin | 27 | 51.4 | 64 | >256 | 33 | 37.6 | 128 | >256 | 18.8 | 80.1 | 2 | >256 | 7.4 | 85.2 | 32 | 128 |
| Imipenem | 72.7 | 18.1 | 8 | 32 | 81.3 | 13.5 | 16 | 32 | 52.5 | 26.5 | 4 | 8 | 61.1 | 23.1 | 4 | 16 |
| Levofloxacin | 78.2 | 16.1 | 32 | 128 | 82 | 12.9 | 32 | 128 | 85.6 | 10.5 | 32 | 64 | 53.7 | 34.3 | 2 | 64 |
| Meropenem | 77 | 17.4 | 32 | 128 | 84.7 | 10.9 | 64 | 256 | 67.4 | 28.7 | 8 | 32 | 58.3 | 25 | 4 | 16 |
| Minocycline | 26.3 | 57.7 | 4 | 32 | 25.6 | 56.3 | 4 | 32 | 32 | 56.4 | 4 | 32 | 28.7 | 59.3 | 4 | 64 |
| Piperacillin/tazobactam | 87 | 9.2 | >256 | >256 | 91 | 7.1 | >256 | >256 | 81.8 | 10.5 | >256 | >256 | 79.6 | 13.9 | 256 | >256 |
| Tigecycline | 2.7 | 91.7 | 0.5 | 2 | 2.9 | 90.2 | 1 | 2 | 1.1 | 98.3 | 0.25 | 1 | 4.6 | 89.8 | 0.5 | 4 |

[a]R, resistant; S, susceptible; CRE, carbapenem-resistant *Enterobacterales*; $MIC_{50/90}$, the MIC ($\mu$g/mL) where 50% or 90% of the isolates were inhibited.

which sequence type 167 (ST167) accounted for 76.9% (20/26) of the isolates, followed by ST746, accounting for 7.7% (2/26). The remaining STs identified were ST405, ST410, ST617, and ST11738, each in one isolate.

Notably, all 26 cefiderocol-resistant *E. coli* isolates harbored NDM-5, and 1 ST167 strain harbored both KPC-2 and NDM-5. AST showed that high-level cefiderocol-resistant isolates (MIC $\geq$32 $\mu$g/mL) accounted for 92.3% (24/26), and only 3 isolates had an MIC of 16 $\mu$g/mL. The MIC values of tigecycline against these 26 isolates were all lower than 1 $\mu$g/mL. The MIC of colistin for all isolates was <0.5 $\mu$g/mL, except for one isolate with an *mcr-1* gene, which exhibited an MIC of 2 $\mu$g/mL.

**TABLE 2** Antimicrobial susceptibility testing results of different carbapenemase-producing CREs[a]

| Antibiotic | KPC-producing isolates ($n = 569$) | | | | NDM-producing isolates ($n = 351$) | | | | IMP-producing isolates ($n = 49$) | | | |
|---|---|---|---|---|---|---|---|---|---|---|---|---|
| | %R | %S | $MIC_{50}$ | $MIC_{90}$ | %R | %S | $MIC_{50}$ | $MIC_{90}$ | %R | %S | $MIC_{50}$ | $MIC_{90}$ |
| Amikacin | 59.1 | 39.7 | 256 | 256 | 18.2 | 79.8 | 4 | 256 | 12.2 | 87.8 | 4 | 256 |
| Aztreonam | 98.6 | 1.4 | >256 | >256 | 68.1 | 24.8 | 64 | >256 | 61.2 | 36.7 | 32 | >256 |
| Aztreonam/avibactam | 0.5 | 99.5 | 0.5 | 1 | 1.1 | 97.2 | 0.125 | 1 | 2 | 98 | 0.125 | 2 |
| Cefepime | 94.9 | 0.9 | 128 | 256 | 97.2 | 0.9 | 128 | >256 | 85.7 | 6.1 | 64 | 256 |
| Cefiderocol | 0.4 | 99.6 | 2 | 2 | 8 | 92 | 2 | 4 | 0 | 100 | 1 | 2 |
| Cefoperazone/sulbactam | 96.1 | 0.7 | >256 | >256 | 98.6 | 0.9 | >256 | >256 | 89.8 | 8.2 | 256 | >256 |
| Cefotaxime | 98.9 | 0.9 | 256 | >256 | 99.7 | 0.3 | >256 | >256 | 93.9 | 6.1 | 256 | >256 |
| Cefoxitin | 93 | 2.8 | 256 | >256 | 98.9 | 0.9 | >256 | >256 | 91.8 | 8.2 | >256 | >256 |
| Ceftazidime | 97.2 | 1.8 | >256 | >256 | 98.9 | 1.1 | >256 | >256 | 93.9 | 6.1 | >256 | >256 |
| Ceftazidime/avibactam | 2.6 | 97.4 | 2 | 4 | 100 | 0 | >256 | >256 | 100 | 0 | >256 | >256 |
| Chloramphenicol | 69.1 | 19.3 | 32 | >256 | 36.5 | 51.6 | 8 | >256 | 28.6 | 59.2 | 8 | 256 |
| Ciprofloxacin | 96 | 2.8 | 64 | 128 | 68.4 | 21.9 | 16 | 128 | 61.2 | 36.7 | 4 | 128 |
| Colistin | 4 | 96 | 0.125 | 0.5 | 2.3 | 97.7 | 0.125 | 0.5 | 0 | 100 | 0.125 | 0.5 |
| Ertapenem | 97.9 | 1.1 | 256 | >256 | 98.9 | 0.6 | 32 | 64 | 79.6 | 14.3 | 8 | 256 |
| Fosfomycin | 39 | 24.8 | 128 | >256 | 12.8 | 83.5 | 16 | 256 | 20.4 | 67.3 | 32 | >256 |
| Imipenem | 88.8 | 7 | 16 | 32 | 74.1 | 7.1 | 4 | 16 | 44.9 | 44.9 | 2 | 32 |
| Levofloxacin | 94.9 | 3.7 | 32 | 128 | 60.4 | 27.4 | 8 | 64 | 59.2 | 36.7 | 8 | 64 |
| Meropenem | 90 | 6.9 | 64 | 256 | 84 | 6.8 | 8 | 32 | 51 | 30.6 | 4 | 64 |
| Minocycline | 20.9 | 59.6 | 4 | 32 | 31.3 | 56.1 | 4 | 32 | 22.4 | 59.2 | 2 | 16 |
| Piperacillin/tazobactam | 97 | 0.9 | >256 | >256 | 95.2 | 2 | >256 | >256 | 59.2 | 36.7 | 256 | >256 |
| Tigecycline | 1.1 | 92.3 | 1 | 2 | 3.7 | 93.4 | 0.5 | 2 | 8.2 | 89.8 | 0.5 | 4 |

[a]R, resistant; S, susceptible; CRE, carbapenem-resistant *Enterobacterales*; $MIC_{50/90}$, the MIC ($\mu$g/mL) where 50 or 90% of the isolates were inhibited.

**TABLE 3** The main characteristics of cefiderocol-resistant *Escherichia coli*[a]

| Isolate no. | Yr | MLST | Carbapenemase | PBP3 mutation | CirA | Antibiotic MIC ($\mu$g/mL) | | | | | |
|---|---|---|---|---|---|---|---|---|---|---|---|
| | | | | | | FDC | MEM | IMP | ETP | COL | TGC |
| C5462 | 2018 | 167 | KPC-2 + NDM-5 | Q227H, 333 aa insertion YRIN, E349K, I532L | Truncated at 109 aa | 128 | 64 | 16 | >256 | ≤0.064 | 1 |
| C5492 | 2019 | 167 | NDM-5 | Q227H, 333 aa insertion YRIN, E349K, I532L | Truncated at 109 aa | 16 | 8 | 8 | 64 | 0.125 | 0.25 |
| C5557 | 2018 | 167 | NDM-5 | Q227H, 333 aa insertion YRIN, E349K, I532L | Truncated at 109 aa | 64 | 16 | 4 | 64 | 0.125 | 0.5 |
| C5655 | 2018 | 167 | NDM-5 | Q227H, 333 aa insertion YRIN, E349K, I532L | Truncated at 109 aa | 128 | 32 | 8 | 128 | 0.125 | 0.5 |
| C5881 | 2019 | 405 | NDM-5 | 333 aa insertion YRIK, A413V | Truncated at 621 aa | 64 | 16 | 4 | 64 | 0.125 | 0.25 |
| C5911 | 2019 | 167 | NDM-5 | Q227H, 333 aa insertion YRIN, E349K, I532L | Truncated at 109 aa | 64 | 8 | 2 | 32 | 0.125 | 0.5 |
| C6242 | 2019 | 167 | NDM-5 | Q227H, 333 aa insertion YRIN, E349K, I532L | Truncated at 109 aa | 64 | 16 | 4 | 32 | 0.25 | 0.25 |
| C6335 | 2018 | 167 | NDM-5 | Q227H, 333 aa insertion YRIN, E349K, I532L | Truncated at 109 aa | 64 | 32 | 4 | 32 | 0.25 | 0.25 |
| C6339 | 2018 | 167 | NDM-5 | Q227H, 333 aa insertion YRIN, E349K, I532L | Truncated at 109 aa | 64 | 16 | 8 | 32 | 0.25 | 0.5 |
| C6340 | 2019 | 167 | NDM-5 | Q227H, 333 aa insertion YRIN, E349K, I532L | Truncated at 109 aa | 64 | 32 | 8 | 64 | 0.25 | 0.25 |
| C6341 | 2019 | 167 | NDM-5 | Q227H, 333 aa insertion YRIN, E349K, I532L | Truncated at 109 aa | 64 | 32 | 4 | 32 | 0.25 | 0.5 |
| C6343 | 2018 | 167 | NDM-5 | Q227H, 333 aa insertion YRIN, E349K, I532L | Truncated at 109 aa | 64 | 32 | 8 | 128 | 2 | 0.5 |
| C6346 | 2018 | 167 | NDM-5 | Q227H, 333 aa insertion YRIN, E349K, I532L | Truncated at 109 aa | 64 | 16 | 4 | 32 | 0.25 | 0.5 |
| C6351 | 2018 | 167 | NDM-5 | Q227H, 333 aa insertion YRIN, E349K, I532L | Truncated at 109 aa | 64 | 16 | 2 | 32 | 0.25 | 0.5 |
| C6352 | 2019 | 410 | NDM-5 | Q227H, 333 aa insertion YRIN, E349K, I532L | WT | 32 | 8 | 2 | 32 | 0.25 | 0.25 |
| C6377 | 2019 | 167 | NDM-5 | Q227H, 333 aa insertion YRIN, E349K, I532L | Truncated at 109 aa | 64 | 16 | 2 | 32 | 0.5 | 0.5 |
| C6422 | 2019 | 167 | NDM-5 | Q227H, 333 aa insertion YRIN, E349K, I532L | Truncated at 109 aa | 16 | 64 | 8 | 128 | 0.25 | 0.5 |
| C6575 | 2019 | 617 | NDM-5 | Q227H, 333 aa insertion YRIN, E349K, I532L | Truncated at 389 aa | 128 | 8 | 8 | 64 | 0.5 | 0.25 |
| C6577 | 2018 | 167 | NDM-5 | Q227H, 333 aa insertion YRIN, E349K, I532L | Truncated at 109 aa | 64 | 32 | 8 | 64 | 0.25 | 0.5 |
| C6579 | 2018 | 167 | NDM-5 | Q227H, 333 aa insertion YRIN, E349K, I532L | Truncated at 109 aa | 64 | 32 | 8 | 64 | 0.25 | 0.5 |
| C6580 | 2018 | 167 | NDM-5 | Q227H, 333 aa insertion YRIN, E349K, I532L | Truncated at 109 aa | 64 | 32 | 8 | 64 | 0.25 | 0.5 |
| C6592 | 2018 | 167 | NDM-5 | Q227H, 333 aa insertion YRIN, E349K, I532L | Truncated at 109 aa | 16 | 8 | 4 | 16 | 0.5 | 1 |
| C6599 | 2019 | 167 | NDM-5 | Q227H, 333 aa insertion YRIN, E349K, I532L | Truncated at 109 aa | 128 | 1 | 1 | 1 | 0.25 | 1 |
| C6617 | 2019 | 11738 | NDM-5 | Q227H, 333 aa insertion YRIN, E349K I532L | WT | 128 | 8 | 4 | 16 | 0.25 | 0.5 |
| C6619 | 2019 | 746 | NDM-5 | Q227H, 333 aa insertion YRIN, E349K, I532L | WT | 64 | 16 | 2 | 32 | 0.25 | 0.5 |
| C6620 | 2019 | 746 | NDM-5 | Q227H, 333 aa insertion YRIN, E349K, I532L | WT | 64 | 16 | 4 | 32 | 0.25 | 0.25 |

[a]FDC, cefiderocol; CAZ, ceftazidime; MEM, meropenem; IMP, imipenem; ETP, ertapenem; COL, colistin; TGC, tigecycline; WT, wild type; aa, amino acids.

Tables S2 and S3 in the supplemental material show the comparison results of all PBPs and TBDTs using protein-BLAST using *E. coli* K-12 as a reference strain. Mutants with four amino acids (YRIN or YRIK) inserted at the 333rd amino acid of PBP3 in all cefiderocol-resistant isolates were observed (Table 3). The TBDTs of *cirA* of the 22 isolates contained a stop codon for the *cirA* gene of the catecholamine siderophore receptor. The TBDT gene encoding CirA has a stop codon in 22 isolates.

A phylogenetic tree based on the core genome showed that the distance of the 26 strains is very similar to that of the MLST. Differences in the resistance genes between ST167 isolates and non-ST167 isolates were observed, especially in extended-spectrum beta-lactamase-related genes ($bla_{CTX-M}$), tetracycline resistance genes (*tet*A, *tet*B), and fosfomycin resistance genes related to resistance (*fos*A) (Fig. 2). As shown in Fig. S1, the phylogenetic tree based on the core genome of 20 strains of ST167 *E. coli* can be seen that the strains from different regions and have large differences between strains. There is no statistically significant difference in the copy number of $bla_{NDM}$ between the resistant and susceptible groups. See Fig. S2 for details.

**Clone transformations and *cirA*, $bla_{NDM}$ gene deletion results.** We constructed a gene expression vector, pEASY-T1-PBP3 (YRIN/YRIK), carrying a mutant copy of PBP3 (YRIN/YRIK) in the cefiderocol-resistant strain (Table 4). After chemical transformation, we introduced the vector into *E. coli* DH5$\alpha$. After IPTG-induced expression, AST showed that the MIC of cefiderocol against DH5$\alpha$ carrying the PBP3 mutant increased by 4-fold (0.25 $\mu$g/mL) compared to its original MIC (0.064 $\mu$g/mL).

We selected two cefiderocol-resistant *E. coli* isolates, C5492 and C6346, for which the MIC values of cefiderocol were 16 $\mu$g/mL and 64 $\mu$g/mL, respectively. The *cirA* genes in these two isolates were premature stop codons. We introduced a wild-type *cirA* gene into these drug-resistant isolates using a vector and induced the expression using IPTG (isopropyl-$\beta$-D-thiogalactopyranoside). AST showed that the MIC of cefiderocol for C5492 and C6346 decreased from 16 to 2 $\mu$g/mL and from 64 to 4 $\mu$g/mL, decreasing 8- and 16-fold, respectively. Therefore, the isolates recovered their

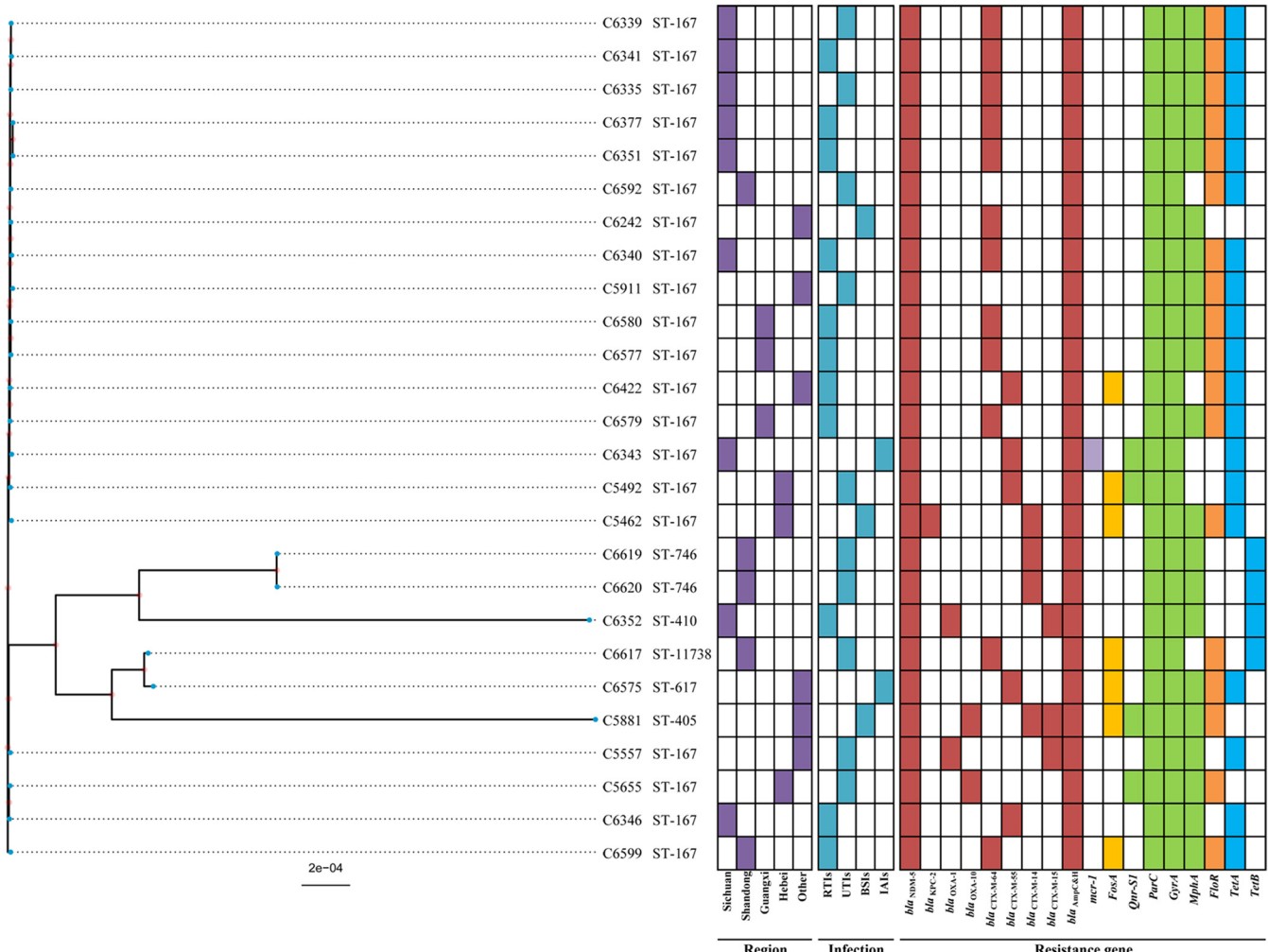

**FIG 2** The phylogenetic tree based on core genomes of 26 cefiderocol-resistant *Escherichia coli* strains and the comparison diagram of detected antimicrobial-resistant genes. The blank part represents the absence of the corresponding gene, and the colored parts represent the presence of the corresponding gene. This figure comprised 3,382/9,018 core to total genes. RTIs, respiratory tract infection; UTIs, urinary tract infection; BSIs, bloodstream infection; IAIs, intra-abdominal infection.

susceptibility after wild-type *cirA* was introduced. We also knocked out the *cirA* and $bla_{NDM}$ genes with the CRISPR-Cas9 genome editing system in a cefiderocol (FDC)-susceptible (MIC = 1 $\mu$g/mL) strain (C7310). It was found that, compared with C7310, the MIC of cefiderocol to C7310$\Delta cirA$ increased by 64 times, from 1 $\mu$g/mL to 64 $\mu$g/mL. When both *cirA* and $bla_{NDM}$ genes were knocked out, the MIC of C7310$\Delta cirA$ $\Delta bla_{NDM}$ decreased from 64 $\mu$g/mL to 0.5 $\mu$g/mL. ATCC 25922 did not increase the MIC of cefiderocol after deletion of the *cirA* gene.

## DISCUSSION

Recent research and development of new antimicrobial agents have provided additional treatment options for CRE infection, especially in European countries and the United States (14). In the United States, the clinical application of new beta-lactam combination agents has resulted in clinical benefits, including reducing mortality related to infections with carbapenem-resistant Gram-negative bacilli (15). In contrast, relatively fewer new anti-CRE drugs have been approved in China. Therefore, there is a great demand for new anti-CRE drugs.

Iron is essential for the survival and metabolism of pathogenic bacteria (16). Siderophores are natural small-molecule compounds produced by bacteria and secreted extracellularly. A transport system transfers the siderophore-$Fe^{3+}$ chelate into the cell to support bacterial

**TABLE 4** Antimicrobial susceptibility profiles of the isolates carrying different clone vectors and *cirA* deletions[a]

| Strain | Description | MLST | Antibiotic MIC ($\mu$g/mL) | | | |
|---|---|---|---|---|---|---|
| | | | FDC | CAZ | MEM | IMP |
| DH5a | *E. coli* Recipient for transformation | | 0.064 | 0.064 | 0.016 | 0.125 |
| DH5a+pEASY-T1 | *E. coli* Transformants | | 0.064 | 0.125 | 0.016 | 0.125 |
| DH5a+pEASY-T1-PBP3 (YRIK) | *E. coli* Transformants | | 0.25 | 0.25 | 0.032 | 0.125 |
| DH5a+pEASY-T1-PBP3 (YRIN) | *E. coli* Transformants | | 0.25 | 0.25 | 0.032 | 0.125 |
| DH5a+pEASY-T1-PBP3 (WT) | *E. coli* Transformants | | 0.064 | 0.125 | 0.016 | 0.125 |
| C5492 | *E. coli* Recipient for transformation | ST167 | 16 | >256 | 8 | 8 |
| C5492+pEASY-T1 | *E. coli* Transformants | ST167 | 16 | >256 | 8 | 8 |
| C5492+pEASY-T1-*cirA*(WT) | *E. coli* Transformants | ST167 | 2 | >256 | 8 | 8 |
| C6346 | *E. coli* Recipient for transformation | ST167 | 64 | >256 | 16 | 4 |
| C6346+pEASY-T1 | *E. coli* Transformants | ST167 | 64 | >256 | 16 | 4 |
| C6346+pEASY-T1-*cirA*(WT) | *E. coli* Transformants | ST167 | 4 | >256 | 16 | 4 |
| C7310 | *E. coli* Cefiderocol-susceptible clinical isolate | ST683 | 1 | >256 | 2 | 2 |
| C7310Δ*cirA* | *E. coli cirA* deletion of C7310 | ST683 | 64 | >256 | 2 | 2 |
| C7310Δ*cirA*Δ*bla*$_{NDM}$ | *E. coli cirA* and *bla*$_{NDM}$ deletion of C7310 | ST683 | 0.5 | 64 | 0.5 | 1 |
| ATCC 25922 | | | 0.125 | 0.125 | ≤0.016 | 0.125 |
| ATCC 25922Δ*cirA* | *E. coli cirA* deletion of ATCC 25922 | | 0.125 | 0.125 | ≤0.016 | 0.125 |

[a]FDC, cefiderocol; CAZ, ceftazidime; MEM, meropenem; IMP, imipenem; WT, wild type.

survival and metabolism (17). Natural siderophores are classified into four types according to functional groups chelated with iron ions: hydroxamic acid, catechol, hydroxycarboxylic acid, and mixed type (18). The siderophore component of cefiderocol is catecholamine.

Our results showed that among the 1,158 CRE isolates tested, the resistance rates of carbapenem-resistant *K. pneumoniae* and *Enterobacter cloaceae* against cefiderocol were 0.3% and 1.9%, respectively. Our results are consistent with the data of many previous studies (10, 11, 15). Surprisingly, 14.4% of carbapenem-resistant *E. coli* were resistant to cefiderocol, which was not reported previously. In 2015, Kohira et al. reported the *in vitro* susceptibility of clinically typical *Enterobacterales* to cefiderocol. Only seven NDM-producing *E. coli* isolates show reduced susceptibility to cefiderocol; however, the resistance mechanism remains unclear (19). The data of SIDERO-WT from 2014 to 2016 show that among the 8,307 isolates of *Enterobacterales* tested, only 44 isolates are resistant to cefiderocol, and most of them are moderately resistant with an MIC value between 8 and 16 $\mu$g/mL; isolates with an MIC of 64 $\mu$g/mL are rare. Our data showed that there are more high-level cefiderocol-resistant isolates among carbapenem-resistant *E. coli* in China. All these isolates harbored NDM-5, which has not been reported before. Previous CRE-Network data reports indicate that NDM-producing *E. coli* is the second-most prevalent CRE strain in China (20, 21). Therefore, determining the resistance mechanism of *E. coli* to cefiderocol may support the rational use of antimicrobial agents, delaying the development of resistance to new drugs. It is quite striking that all cefiderocol-resistant *E. coli* isolates harbored NDM. Nurjadi et al. have reported that NDM facilitated the emergence of cefiderocol resistance in *E. cloacae* (22). Therefore, we should be alert to the high risk of resistance to cefiderocol in the treatment of NDM-producing CRE.

PBPs are essential enzymes in cell wall peptidoglycan synthesis and are also important targets for beta-lactam drugs (23). In 2017, Ito et al. selected typical ATCC isolates and tested the affinity of cefiderocol for different PBPs of *E. coli*. They found that among *E. coli* PBPs, cefiderocol had the highest affinity for PBP3, with an 50% inhibitory concentration (IC$_{50}$) of 0.04 mg/L, followed by PBP2 with an IC$_{50}$ of 2.12 mg/L (24). In 2015, Alm et al. reported that the 333rd position of PBP3 in an NDM-producing *E. coli* strain contained YRIN or YRIK, a four-amino acid insertion associated with aztreonam-avibactam resistance (25). Similarly, a point mutation in *Acinetobacter baumannii*-derived PBP3 may cause cefiderocol resistance (26). The results of our cloning transformation showed that the MIC of cefiderocol against DH5$\alpha$ containing a PBP3 mutation (YRIN or YRIK) increased by 4-fold; however, it did not reach the resistance level. Mutations in PBP3 do not considerably affect cefiderocol resistance significantly. This result demonstrates that the PBP3 mutation may not serve as an efficient determinant of cefiderocol resistance and is only an auxiliary effect.

An essential issue in the Trojan Horse strategy is if the trojan horse is unable to enter the city. Most pathogenic bacteria have multiple approaches for acquiring iron, and the more genes involved in iron transport, the higher the possibility of the bacteria developing drug resistance. Currently, seven TBDTs in *E. coli* rely on the TonB system, namely, FepA, FecA, FhuA, CirA, Fiu, BtuB, and FhuE. FepA, CirA, and Fiu are mainly responsible for transporting catechol-type siderophores (17, 27). In our study, 23 cefiderocol-resistant *E. coli* isolates had *cirA* gene termination codes. After obtaining a wild-type *cirA* gene and inducing expression, the MIC of cefiderocol against the transformant strain decreased to the susceptible range ($\leq$4 $\mu$g/mL); the strain also carried the PBP3 mutation. This result confirmed that the truncation of *cirA* may be the main reason for cefiderocol resistance. Ito et al. showed that the MIC of cefiderocol against the *E. coli* K-12 strain increases by 16-fold when the iron transporter *cirA* and *fiu* genes are knocked out simultaneously (24). Klein et al. reported that rapid development of cefiderocol resistance in *E. cloacae* during treatment is associated with heterogeneous mutations in the catechin siderophore receptor *cirA* (28). Unlike this study, our resistant strains were not subjected to antibiotic pressure from cefiderocol. Kohira et al. reported that beta-lactamase PER is associated with cefiderocol resistance, and our study did not find such extended-spectrum beta-lactamase (ESBL) genes (29). Simner et al. also recently reported an increase in $bla_{NDM}$ copy number under antibiotic pressure, resulting in high expression of NDM, leading to cefiderocol resistance (30). Our data show that the presence of the $bla_{NDM}$ gene in *cirA* knockout strains plays an important role in cefiderocol resistance.

This study has a few limitations. (i) Although our study is a multicenter study, it is not comprehensive enough at the regional and hospital level, and there is still some bias in the data. Follow-up studies with larger and more comprehensive sample sizes are needed. (ii) We could not determine the entire resistance mechanism of the four wild-type *cirA*-encoding cefiderocol-resistant *E. coli* isolates. (iii) Also, non-*E. coli* resistant isolates were not investigated. The resistance mechanism of *K. pneumoniae* and the *Enterobacter cloacae* complex still needs follow-up research and exploration.

To the best of our knowledge, this is the first large-scale *in vitro* AST study of CRE isolates against cefiderocol and other drugs in China. Carbapenem-resistant *E. coli* shows resistance to cefiderocol by the termination of *cirA* coding combined with PBP3 insertion mutation. This study provides a reference for the application of cefiderocol in China.

## MATERIALS AND METHODS

**Bacterial isolates.** From January 2018 to December 2019, 1,158 CRE isolates were collected from 48 hospitals in 23 provinces and municipalities across China, including 38 tertiary hospitals and 10 secondary hospitals, as part of the China CRE-Network research (20). The definition of CRE refers to the definition published by the U.S. CDC (https://www.cdc.gov/hai/organisms/cre/technical-info.html#Definition). The infection types of the isolates include respiratory tract infection, accounting for 41.6% (482/1158), intra-abdominal infection, accounting for 13.4% (155/1158), urinary tract infection, accounting for 13.3% (154/1158), bloodstream infection, accounting for 13.1% (152/1158), and other infection type, accounting for 18.6% (215/1158). CRE were defined as members of the *Enterobacterales* resistant to imipenem, meropenem, ertapenem, or doripenem, or any one of the carbapenems, or producers of carbapenemase based on laboratory experiments, such as modified carbapenem inactivation method (mCIM) according to CLSI (31). The species of all *Enterobacterales* were reconfirmed according to matrix-assisted laser desorption ionization–time of flight mass spectrometry (Bruker Daltonik, Bremen, Germany) at a central laboratory (Peking University People's Hospital) and were stored at −80℃ for further use.

**Antimicrobial susceptibility testing.** Antimicrobial susceptibility testing was performed via the agar dilution and broth microdilution methods at Peking University People's Hospital according to the Clinical and Laboratory Standards Institute (CLSI) guideline (32), and the results were interpreted according to CLSI M100, 30th edition, categories and MIC breakpoints (33). The MIC of cefiderocol was determined using iron-depleted cation-adjusted Mueller Hinton broth (ID-CAMHB) as described in CLSI M100, 30th edition, and previous studies (34). Susceptibility to aztreonam, aztreonam/avibactam, cefoxitin, cefotaxime, ceftazidime, ceftazidime/avibactam, cefepime, piperacillin/tazobactam, cefoperazone/sulbactam, fosfomycin, ertapenem, imipenem, meropenem, amikacin, ciprofloxacin, minocycline, chloramphenicol, and levofloxacin was tested via the agar dilution method. Tigecycline and colistin susceptibility was tested via the broth microdilution method (32). The breakpoint of tigecycline for *Enterobacterales* was obtained from the U.S. FDA standard (https://www.fda.gov/drugs/development-resources/tigecycline-injection-products). The breakpoints of colistin were used in the breakpoint tables to interpret the MIC version 11.0 published by the European Committee on AST. *Pseudomonas aeruginosa* ATCC 27853 and *Escherichia coli* ATCC 25922 were used as

quality control standards for AST. Susceptibility data were analyzed using WHONET v5.6 (http://www.whonet.org/contact.html).

**Investigation of carbapenem resistance mechanisms.** All 1,158 isolates were subjected to mCIM to determine whether carbapenemase is phenotypically produced. PCR analysis was used to detect the genes encoding carbapenemases ($bla_{KPC}$, $bla_{NDM}$, $bla_{IMP}$, $bla_{VIM}$, $bla_{SIM}$, and $bla_{OXA-48}$) as previously described (20, 35, 36). PCR products were purified using a QIAquick PCR purification kit (Qiagen, Valencia, CA, USA) and sequenced via Sanger sequencing on an ABI PRISM 3730XL system (Applied Biosystems, Foster City, CA, USA). The full-length sequence obtained was submitted to the Beta-Lactamase Database (BLDB) for comparison and analysis to obtain the carbapenemase genotype (37).

**Whole-genome sequencing, assembly, and annotation.** A total of 26 cefiderocol-resistant CRE were subjected to whole-genome sequencing. Total DNA was extracted using the TIANamp bacterial DNA kit DP302 (Tiangen Biotech, Beijing, China), and genomic DNA was sequenced using the Illumina NextSeq 550 platform, which produced 150-bp paired-end reads and at least 100-fold coverage of raw reads. The short-read sequence was assembled de novo using SPAdes v3.10.0 (38). The resulting assemblies were annotated using Prokka v1.12 (39), and the core and accessory genomes were defined and extracted using Roary v3.11.2 (40). We constructed a phylogenetic tree in RAxML by using a general time reversible (GTR) model and 1,000 bootstrap replicates (41). All resistance genes, including beta-lactamase and colistin resistance (mcr-1) and other resistance genes, were detected using ResFinder (https://cge.cbs.dtu.dk/services/ResFinder/) and the Basic Local Alignment Search Tool (BLAST). We selected 18 ST167 cefiderocol-susceptible NDM-5 E. coli strains that had previously undergone WGS as the control group and 20 ST167 NDM-5 producing strains in the cefiderocol-resistant group for $bla_{NDM}$ copy number comparison. $bla_{NDM}$ copy numbers were estimated by mapping the raw short-read data to our short-read de novo contig assemblies using the Burrows-Wheeler Aligner (BWA) v3.1 (42). The absolute number of reads that mapped to the $bla_{NDM}$ gene, normalized by sequence depth, was used to generate a sequence depth of coverage.

**Subcloning experiments.** Two different recombinant plasmids were constructed to determine the impact of a pbp3 gene sequence containing a 12-bp insertion and truncated cirA gene on cefiderocol resistance. The gene sequences for the two amino acid insertions of PBP3 were amplified from C5881 (PBP3: YRIK) and C5492 (PBP3: YRIN). A cirA wild-type gene sequence was amplified from E. coli DH5α (derived from E. coli K-12). The plasmid pEASY-T1 was used as a template to amplify all sequences except the coding sequence of lacZ to retain the promoter region of lacZ. The primers and PCR cycling conditions used are listed in Table S1. PCR amplification was performed using a PrimeSTAR high-sensitivity (HS) kit (TaKaRa Biomedical Technology, Beijing, China). The vector PCR product and insertion PCR product were ligated using NEBuilder high-fidelity (HiFi) DNA assembly master mix (New England Biolabs, Ipswich, MA, USA). The assembly reagent (2 μL) containing recombinant plasmids pEASY-T1-PBP3 (YRIK), pEASY-T1-PBP3 (YRIN), pEASY-T1-PBP3 (W.T.), and pEASY-T1-CirA (W.T.) were added to competent E. coli DH5α (100 μL) via incubation on ice for 30 min followed by incubation for 45 s at 42°C. Subsequently, 1 mL of prewarmed SOC medium (37°C; TaKaRa Biomedical Technology) was added to the competent cells. The solution was incubated for 1 h at 37°C with shaking at 200 rpm, and then 100 μL of the bacterial solution was inoculated onto Luria-Bertani (LB) agar containing 100 μg/mL of ampicillin following overnight incubation. Positive clones were selected for subsequent PCR to verify the success of the transformation. E. coli DH5α carrying plasmid W.T. was enriched in LB broth containing 100 μg/mL of ampicillin at 37°C overnight. For plasmid extraction, the bacterial solution (1 mL) was processed using a plasmid minikit (Omega Bio-tek, Norcross, GA, USA). Subsequently, 50 ng of DNA was transformed into cefiderocol-resistant isolates C5492 and C6346. The transformation was performed via electroporation using the MicroPulser electroporator (Bio-Rad, Hercules, CA, USA) using the program EC3 (3.0 kV, 5.5 ms). The cells were then immediately treated with 1 mL of SOC culture (preheated at 37°C) for 1 h. Then, 100 μL of the bacterial solution was inoculated onto LB agar containing 100 μg/mL of kanamycin following overnight incubation. Positive clones were selected for subsequent PCR analysis to verify the success of the transformation. All transformants and parental isolates were subjected to AST for cefiderocol under induction using 1 mM isopropyl beta-D-1-thiogalactopyranoside (IPTG).

**cirA and $bla_{NDM}$ gene deletion.** The E. coli cirA and $bla_{NDM}$ gene deletion strain was obtained from E. coli ATCC 25922 and a clinically derived carbapenem-resistant E. coli C7310 by the CRISPR-Cas9 genome editing system according to methods described previously (43, 44). C7310 was a cefiderocol-susceptible (MIC = 1 μg/mL) E. coli harbored NDM-5 with a wild-type cirA gene. Briefly, the pCas plasmid (#62225) carrying kanamycin resistance was transformed into ATCC 25922 and C7310 strains using electroporation. The colonies were selected on an LB agar plate containing 50 μg/mL kanamycin at 30°C. Annealed cirA or $bla_{NDM}$ spacer oligonucleotides and the repair arms of the cirA gene (~1 kb) were inserted into the pSGKP-spe plasmid (#117234) by Golden Gate assembly (New England Biolabs, Ipswich, MA, USA). Then, the cirA spacer with the repair arms of the cirA gene (~1 kb) and introduced pSGKP-spe plasmid was transformed into the ATCC 25922 and C7310 strains with pCas plasmid using electroporation. The colonies were selected on an LB agar plate containing 100 μg/mL spectinomycin and 50 μg/mL kanamycin at 30°C with 0.2% L-arabinose. The successful cirA deletion strain was confirmed by PCR and sequencing. After confirmation, the pCas and pSGKP-spe plasmids were cured by culturing the cells with at 37°C and in the presence of sucrose. The $bla_{NDM}$ gene deletion was performed based on C7310ΔcirA, following the same steps as described above. For information on primers, please refer to Table S1.

**Data availability.** The complete sequences of the 26 cefiderocol-resistant Escherichia coli isolates and assembled data have been deposited on NCBI with BioProject no. PRJNA756960 (Table S4).

## SUPPLEMENTAL MATERIAL

Supplemental material is available online only.

**SUPPLEMENTAL FILE 1**, PDF file, 0.5 MB.

## ACKNOWLEDGMENTS

We thank the China CRE-Network laboratories.

This study was partly supported by the National Natural Science Foundation of China (no. 32141001, and no. 82172310), the Beijing Natural Science Foundation (no. 7222203), and the Research and Development Fund of Peking University People's Hospital (no. RS2020-02).

We have no transparency declarations.

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
