## [Reviewer comments · Microbiology Spectrum]

Microbiology Spectrum

Occurrence of high levels cefiderocol resistance in carbapenem-resistant *Escherichia coli* before its approval in China: A report from China CRE-Network

Qi Wang, Longyang Jin, Shijun Sun, Yuyao Yin, Ruobing Wang, Fengning Chen, Xiaojuan Wang, Yawei Zhang, Jun Hou, Yumei Zhang, Zhijie Zhang, Liuchun Luo, Zhu Guo, Zhenpeng Li, Xin Lin, Lei Bi, and Hui Wang

Corresponding Author(s): Hui Wang, Institute of Medical Technology, Peking University Health Science Center

Review Timeline:

Submission Date:	December 17, 2021
Editorial Decision:	February 14, 2022
Revision Received:	April 8, 2022
Accepted:	April 14, 2022

Editor: Daria Van Tyne

Reviewer(s): The reviewers have opted to remain anonymous.

Transaction Report:

DOI: <https://doi.org/10.1128/spectrum.02670-21>

February 14, 2022

Prof. Hui Wang
Department of Clinical Laboratory, Peking University People's Hospital, Beijing, China
Xueyuan Road, No.38 Beijing 100191, China
Beijing 100191
China

Re: Spectrum02670-21 (Emergence of high levels cefiderocol resistance in carbapenem-resistant Escherichia coli before its approval in China: A report from China CRE-Network)

Dear Prof. Hui Wang:

Thank you for submitting your manuscript to Microbiology Spectrum. Your manuscript has been reviewed by three experts, and I would now like you revise your study in line with their feedback. In addition to the instructions below, please also add a "Data Availability Statement" to the Methods section of your manuscript. More information is available here:
<https://journals.asm.org/open-data-policy>

Link Not Available

Sincerely,

Daria Van Tyne

Journals Department
Reviewer comments:

Reviewer #1 (Comments for the Author):

This manuscript systematically describes the epidemiological status of CRE in China and investigated the drug resistance mechanism of 26 FDC resistant E. coli. The overall content is comprehensive, although there is a lack of new findings, and both PB3 mutation and cirA truncation are not new.
1, the reconfirmation of CRE with MALDI-tof is unreasonable, which can only be used as bacterial specie identification(line 115-

- 118);
2. It should be noted that the drug list (in the first column) in Table 1 and 2 is neither arranged in alphabetical order nor in drug classification.
 3. It is better to conduct the study on the impact of *cirA* deletion on sensitivity to FDC.

Reviewer #2 (Comments for the Author):

This study by Wang and colleagues reports cefiderocol susceptibility of CRE clinical isolates collected from hospitals across China and attempt to determine the resistance mechanisms in *E. coli*. 1158 isolates were tested, and aztreonam-avibactam and cefiderocol showed the highest activity. Cefiderocol resistance was relatively common in *E. coli* accounting for 26 of 30 resistant isolates, therefore these *E. coli* isolates were investigated further. WGS showed that they all carried bla_{NDM-5}, and ST167 accounted for the majority. 22 of 26 isolates had four-amino-acid insertions (YRIN/YRIK) at position 333 of PBP3 previously associated with cefiderocol resistance, and 22 isolates had truncation of siderophore receptor CirA. The impact of these genetic changes on the cefiderocol MICs were verified by complementation of the respective genes. The results showed that loss of functional CirA was impactful, whereas the PBP3 insertion resulted in a modest four-fold MIC increase. These results are more or less expected, but some elements (*E. coli* having the highest cefiderocol resistance, with resistance concentrated in ST167) are new and merit attention. The approach to the study is generally sound.

Comments for the authors' consideration:

- Why were only *E. coli* investigated and not the two *K. pneumoniae* and the two *E. cloacae*?
- For the YRIN/YRIK complementation experiment, a negative control (complementation with wildtype PBP3) is required to rule out a dose-response effect as the cause of the observed MIC change.
- Line 47: premature stop codon
- Line 49: in which
- Line 51: "high activity" is confusing. "active against most..." would be more accurate.
- Line 53: expression in outer membrane was not confirmed in this study
- Line 71: "efficient" is vague - would rewrite. E.g. "not always active", "difficult to use"
- Line 75: strains that produce metallo-beta-lactamase
- Line 115: according to
- Line 188: What are the differences between pEASY-T1-PBP3 (YRIK) and pEASY-T1-PBP3 (YRIK)?
- Line 213: How was *Enterobacter cloacae* identified?
- Line 259: Is ST11738 correct - that's a big number...
- Line 270: It is unclear whether 26 or 22 isolates had a stop codon in *cirA*. Also, where in the gene this occurred can be summarized in the text.
- Line 274: a phylogenetic tree based on the core genome. I don't think evolution can be inferred in this setting. Also, what is homology?
- Line 279: Figure S1
- Line 294: How was the lack of *cirA* expression confirmed?
- Line 296: decreased
- Line 356: mutation
- Line 356: truncation of CirA
- Line 364: What is cefpiryl?
- Line 369: Also, non-*E. coli* resistant isolates were not investigated.

Reviewer #3 (Comments for the Author):

The work of Wang et al. investigated over 1000 isolates from the CRE- Network in China regarding phenotypical resistance toward several antibiotics including cefiderocol and molecular characteristics of the carbapenem-resistant strains. Special emphasis is put on the cefiderocol-resistant isolates as they have been further investigated by whole genome sequencing. The authors describe mutations in the *cirA* and *pbp3* genes that are related to elevated MICs and resistance to cefiderocol. The study provides interesting insight into the resistance patterns of CRE, distribution of carbapenemases and the resistance mechanism of cefiderocol-resistant *E. coli* isolates in China. The methods used are appropriate and I think it provides valuable information.

The manuscript would benefit from some modifications and more detailed description at several points.

1. The title needs to be changed. I think it is not appropriate to talk about emergence when there is no data available that would suggest a rise of resistance rates. I would rather recommend to use the term occurrence or anything comparable that is more neutral.
2. How were the bacterial isolates selected for the CRE-network? Is that a network that includes all hospitals in China or special types of hospitals (e.g. University, tertiary care)? Were there special patient cohorts selected for the study? Do this cover any type of infection or just the ones mentioned in the data? Please be more specific so that it is clear how the isolates were selected or if there was any type of selection.
3. I was unable to find the sequencing statistics. This needs to be included in the supplementary material. Please state how

- many genes were in the core genome. Please also include the accession numbers of the sequences as supplementary material.
4. 1003/1158 strains were positive in the mCIM. 990/1158 isolates were positive in the PCR for the most frequently encountered carbapenemases. How do they overlap, meaning how many of the mCIM positive isolates were positive in the PCR? Were there also mCIM positive with negative PCR and vice versa?
 5. A PCR for the most common carbapenemase genes was applied. What about the CRE negative in your multiplex PCR with positive mCIM (if there were any)? Were they sequenced to detect carbapenemases that were not in the PCR panel?
 6. Line 262 and table 3: Did you determine the exact MIC of cefiderocol for the isolates with MIC >64mg/L? This would be very interesting.
 7. Line 272; *cirA* is a catecholamine siderophore receptor.
 8. It is quite striking that all cefiderocol resistant *E. coli* isolates harbored NDM. I think this can be even more highlighted in the discussion as this is relevant for therapeutic decisions. The role of NDM in the resistance towards cefiderocol has been described for *E. cloacae* recently (Nurjadi et al. New Delhi metallo-beta-lactamase facilitates the emergence of cefiderocol resistance in *Enterobacter cloacae*. Antimicrob Agents Chemother. 2021 Dec 6: AAC0201121. doi: 10.1128/AAC.02011-21), this may be included in the discussion as it seems to be an effect of the type of the β -lactamase which may be independent of the bacterial species.
 9. The limitations sections needs to be expanded. Please be more critical. Was there bias in sampling and is the data representative for China? Do your experiments on *cirA* and *pbp3* really confirm the resistance mechanism to cefiderocol? Etc.
 10. Tables 1 and 2, line 527. MIC 50, MIC 90, this refers to the MIC where 50 or 90% of the isolates were inhibited. Please describe correctly. Why is there a line empty before cefotixime in table 1 and before cefoperazone/sulbactam in table 2?
 11. All tables with MIC values: please mention the unity (mg/L or μ g/ml as I suppose).
 12. Please expand all figure legends/captions.
 - a. Figure 1: change title to "Distribution of the most frequent carbapenemases carried...". Legend: green: "no carbapenemase detected by PCR". Please explain the figure to a minimal extent in the caption.
 - b. Figure 2: Expand the legend to the meaning of the colors, does the color mean the gene is present or absent? The genes need to be named in the legend as well. The scale for the genomic distance seems to be truncated and the unit is missing in the picture. I would further suggest to change the expression "comparison diagram..." to "detected antimicrobial resistance genes".

Minor comments:

Line 42: „aztreonam-avibactam had..."

Line 48: "..., the MIC of the transformants

Line 50: delete "was"

Line 71-72 Indeed tigecycline is not useful for all types of infection, especially not for blood stream infections. However, intraabdominal infections and especially those affecting the bile duct system can be treated successfully as the concentration in bile is comparably high. Please rephrase the sentence in your manuscript, as it is not completely correct as it is written.

Line 72: colistin also has a neurotoxicity which is clinically relevant. Please insert this information.

Line 75: please change to "... it is ineffective against metallo- β -lactamase bearing gram-negative bacilli".

Line 80: Please cite a work of Marvin J Miller when referring to his work and group. There's a bunch of publications.

Line 99: I do not agree that it is possible to "eradicate" CRE. It is a promising treatment option. Please rephrase.

Lines 193 and 204: What is an "appropriate amount"? Please specify.

Line 219: is suppose you mean these isolates had a positive result in the PCR for carbapenemases.

Line 221: Please change "produced" to detected. And lines 225 and following: change "produced" to harbored or carried.

Line 271: please change to "...contains a stop codon." Please also consider changing the term stopped encoding to a stop coding or any other expression that is commonly used.

Line 274 ff.: Phylogenetic tree is commonly used as expression here, while evolutionary tree is an older- however still correct-expression.

Line 279: Figure instead of Figura

Line 298, susceptibility instead of sensitivity

Line 347: I think you mean "An essential issue in the Trojan Horse strategy is if the trojan horse is unable to enter the city"

Line 360, following "Klein et al." This sentence does not make sense as it is written, please rephrase.

Line 360 and following, it is more common to not give the full name, instead put Klein et al. etc. for the cited authors/group.

Staff Comments:

Preparing Revision Guidelines

- Point-by-point responses to the issues raised by the reviewers in a file named "Response to Reviewers," NOT IN YOUR COVER LETTER.

- Upload a compare copy of the manuscript (without figures) as a "Marked-Up Manuscript" file.
- Each figure must be uploaded as a separate file, and any multipanel figures must be assembled into one file.
- Manuscript: A .DOC version of the revised manuscript
- Figures: Editable, high-resolution, individual figure files are required at revision, TIFF or EPS files are preferred

Please return the manuscript within 60 days; if you cannot complete the modification within this time period, please contact me. If you do not wish to modify the manuscript and prefer to submit it to another journal, please notify me of your decision immediately so that the manuscript may be formally withdrawn from consideration by Microbiology Spectrum.

Response to Reviewers

We would like to thank the Editor for the chance to review the manuscript and wish to thank the reviewer for their thorough assessment and useful comments. We have implemented essentially all of the suggestions.

A point-by-point response to all comments is included below. Our responses are shown in red immediately under each of the reviewer's comments.

We hope that with these changes the manuscript is now suitable for publication. Thank you for your time and consideration.

Yours sincerely,

Hui Wang, on behalf of all the co-authors

Reviewer comments:

Reviewer #1 (Comments for the Author):

This manuscript systematically describes the epidemiological status of CRE in China and investigated the drug resistance mechanism of 26 FDC resistant *E. coli*. The overall content is comprehensive, although there is a lack of new findings, and both PBP3 mutation and *cirA* truncation are not new.

Thank you.

1. the reconfirmation of CRE with MALDI-tof is unreasonable, which can only be used as bacterial specie identification (line 115-118);

We only used MALDI-TOF to identify the Enterobacterales species. The description of CRE is not accurate enough. We have revised the original text. Please see lines 130-134 in the clean version.

2. It should be noted that the drug list (in the first column) in Table 1 and 2 is neither arranged in alphabetical order nor in drug classification.

Thanks, we have reordered the antimicrobials alphabetically. Please see Table 1 and Table 2 in the clean version.

3. It is better to conduct the study on the impact of *cirA* deletion on sensitivity to FDC.

Thank you. We knocked out the *cirA* gene by the CRISPR-Cas9 genome editing system in an FDC-susceptible (MIC=1 µg/ml) *E. coli* (C7310) and ATCC25922. Compared with C7310, the MIC of C7310Δ*cirA* to cefiderocol increased by 64 times, from 1 µg/ml to 64 µg/ml. ATCC25922 did not increase the MIC of cefiderocol after deletion of the *cirA* gene. We hypothesized that it may be related to the existence of NDM. Therefore, we delete the *bla*_{NDM} gene of C7310Δ*cirA*. The MIC of C7310Δ*cirA*Δ*bla*_{NDM} to cefiderocol was changed from 64 µg/ml to 0.5 µg/ml. Please see lines 329-335 and Table 4 in the clean version.

Reviewer #2 (Comments for the Author):

This study by Wang and colleagues reports cefiderocol susceptibility of CRE clinical isolates collected from hospitals across China and attempt to determine the resistance mechanisms in *E. coli*. 1158 isolates were tested, and aztreonam-avibactam and cefiderocol showed the highest activity. Cefiderocol resistance was relatively common in *E. coli* accounting for 26 of 30 resistant isolates, therefore these *E. coli* isolates were investigated further. WGS showed that they all carried bla_{NDM-5}, and ST167 accounted for the majority. 22 of 26 isolates had four-amino-acid insertions (YRIN/YRIK) at position 333 of PBP3 previously associated with cefiderocol resistance, and 22 isolates had truncation of siderophore receptor CirA. The impact of these genetic changes on the cefiderocol MICs were verified by complementation of the respective genes. The results showed that loss of functional CirA was impactful, whereas the PBP3 insertion resulted in a modest four-fold MIC increase. These results are more or less expected, but some elements (*E. coli* having the highest cefiderocol resistance, with resistance concentrated in ST167) are new and merit attention. The approach to the study is generally sound.

Thank you.

Comments for the authors' consideration:

-Why were only *E. coli* investigated and not the two *K. pneumoniae* and the two *E. cloacae*?

Thanks for your question. Our results showed that among the 1,158 CRE isolates tested, the resistance rates of carbapenem-resistant *K. pneumoniae* and *E. cloacae* against cefiderocol were only 0.3% and 1.9%, respectively. *E. coli* has a high resistance rate (14.3%) to cefiderocol, rare in previous reports. Therefore, we focused our research on the resistance of *E. coli*. We will continue to focus on *K. pneumoniae* and *E. cloacae* complex in future studies.

-For the YRIN/YRIK complementation experiment, a negative control (complementation with wildtype PBP3) is required to rule out a dose-response effect as the cause of the observed MIC change.

Thanks. We have added a negative control (complementation with wildtype PBP3) to rule out a dose-response effect in the YRIN/YRIK complementation experiment. Please see the Table 4 in the clean version.

-Line 47: premature stop codon

Thanks, it has been edited. Please see line 46 in the clean version.

-Line 49: in which

Corrected. Please see line 50 in the clean version.

-Line 51: "high activity" is confusing. "active against most..." would be more accurate.

Thanks, it has been edited. Please see line 51 in the clean version.

-Line 53: expression in outer membrane was not confirmed in this study

Thanks, it has been edited. Please see line 53 in the clean version.

-Line 71: "efficient" is vague - would rewrite. E.g. "not always active", "difficult to use"

Thanks, it has been edited. Please see line 81 in the clean version.

-Line 75: strains that produce metallo-beta-lactamase

Thanks, it has been edited. Please see line 88 in the clean version.

-Line 115: according to

Thanks, it has been edited. Please see line 130 in the clean version.

-Line 188: What are the differences between pEASY-T1-PBP3 (YRIK) and pEASY-T1-PBP3 (YRIK)?

It's a typo, and it should be pEASY-T1-PBP3 (YRIK) and pEASY-T1-PBP3 (YRIN). Thanks, it has been edited. Please see line 190 in the clean version.

-Line 213: How was *Enterobacter cloacae* identified?

Thanks. MALD-TOF could not fully distinguish different subspecies in the *Enterobacter cloacae* complex. Therefore, we use the term *Enterobacter cloacae* complex instead of *Enterobacter cloacae*. We have changed the term of *Enterobacter cloacae* throughout the text.

-Line 259: Is ST11738 correct - that's a big number...

Thanks. ST11738 has been confirmed.

-Line 270: It is unclear whether 26 or 22 isolates had a stop codon in *cirA*. Also, where in the gene this occurred can be summarized in the text.

We have corrected it, and it should be 22 strains that have the premature termination of the *cirA* gene.

-Line 274: a phylogenetic tree based on the core genome. I don't think evolution can be inferred in this setting. Also, what is homology?

Thanks for your comment. A phylogenetic tree like Table 2 does not show evolutionary relationships. Only the relationships between strains in the phylogenetic tree can be shown. We have revised the relevant description. Please see lines 306-314 in the clean version.

-Line 279: Figure S1

Thanks, it has been edited.

-Line 294: How was the lack of *cirA* expression confirmed?

Thanks, the description here is not accurate enough. It should be *cirA* gene premature stop codon. It has been edited.

-Line 296: decreased

Thanks, it has been edited. Please see line 327 in the clean version.

-Line 356: mutation

Thanks, it has been edited. Please see line 395 in the clean version.

-Line 356: truncation of CirA

Thanks, it has been edited. Please see line 395 in the clean version.

-Line 364: What is cefpiryl?

Corrected. Please see line 402 in the clean version.

-Line 369: Also, non-E. coli resistant isolates were not investigated.

Thanks, we have added this sentence to the study limitations. Please see line 414-418 in the clean version.

Reviewer #3 (Comments for the Author):

The work of Wang et al. investigated over 1000 isolates from the CRE- Network in China regarding phenotypical resistance toward several antibiotics including cefiderocol and molecular characteristics of the carbapenem-resistant strains. Special emphasis is put on the cefiderocol-resistant isolates as they have been further investigated by whole genome sequencing. The authors describe mutations in the *cirA* and *pbp3* genes that are related to elevated MICs and resistance to cefiderocol. The study provides interesting insight into the resistance patterns of CRE, distribution of carbapenemases and the resistance mechanism of cefiderocol-resistant *E. coli* isolates in China. The methods used are appropriate and I think it provides valuable information.

Thank you.

The manuscript would benefit from some modifications und more detailed description at several points.

1. The title needs to be changed. I think it is not appropriate to talk about emergence when there is no data available that would suggest a rise of resistance rates. I would rather recommend to use the term occurrence or anything comparable that is more neutral.

Thanks for your very professional suggestion. We have revised the title according to your suggestion. Please see the title in the clean version.

2. How were the bacterial isolates selected for the CRE-network? Is that a network that includes all hospitals in China or special types of hospitals (e.g. University, tertiary care)? Were there special patient cohorts selected for the study? Do this cover any type of infection or just the ones mentioned in the data? Please be more specific so that it is clear how the isolates were selected or if there was any type of selection.

CRE-network has been a multi-center epidemiological investigation project of carbapenem-resistant Enterobacteriales in China since 2012 (PMID: 30423057). This study included all CRE strains in the CRE-network in 2018-2019. In terms of hospitals, there are 48 hospitals in 23 provinces and cities, including 38 tertiary hospitals and 10 secondary hospitals. The infection types of the isolated strains include respiratory tract infection accounting for 41.6% (482/1158), intra-abdominal infection accounting for 13.4% (155/1158), urinary tract infection accounting for 13.3% (154/1158), bloodstream infection accounting for 13.3% (154/1158) 13.1% (152/1158) and other infection type accounted for 18.6% (215/1158). We have added the description of this section to the original text. Please see line 122-126 in the clean version.

3. I was unable to find the sequencing statistics. This needs to be included in the supplementary material. Please state how many genes were in the core genome. Please also include the accession numbers of the sequences as supplementary material.

Thanks. We have added the sequencing statistics as supplementary material which including species, bioproject number (NCBI), accession number (NCBI), contig_count, N50, largest_contig, total_size. Please see Table S4. The numbers of core genes and the number of total genes have been added to the figure captions in Figure 2 and Figure S2. Figure 2 comprised 3382/9018 core to total genes. Figure S2 comprised 4228/6533 core to total genes.

4. 1003/1158 strains were positive in the mCIM. 990/1158 isolates were positive in the PCR for the most frequently encountered carbapenemases. How do they overlap, meaning how many of the mCIM positive isolates were positive in the PCR? Were there also mCIM positive with negative PCR and vice versa?

The 990 PCR-positive strains were included in the 1003 mCIM-positive strains. Thus, there were 13 mCIM-positive but PCR-negative strains.

5. A PCR for the most common carbapenemase genes was applied. What about the CRE negative in your multiplex PCR with positive mCIM (if there were any)? Were they sequenced to detect carbapenemases that were not in the PCR panel?

A total of 13 mCIM-positive but PCR-negative strains were found in this study. So far we have not tested it for other carbapenemase genes in the PCR panel. We plan to perform next-generation sequencing in the future to explore whether there are other resistance genes and new resistance genes.

6. Line 262 and table 3: Did you determine the exact MIC of cefiderocol for the isolates with MIC >64mg/L? This would be very interesting.

Thanks, we have performed AST on four strains with MIC>64 mg/L, and the results showed that they were all 128 mg/L. We have updated the data in the original text. Please see Table 3 in the clean version.

7. Line 272; cirA is a catecholamine siderophore receptor.

Thanks, it has been edited. Please see line 304 in the clean version.

8. It is quite striking that all cefiderocol resistant E.coli isolates harbored NDM. I think this can be even more highlighted in the discussion as this is relevant for therapeutic decisions. The role of NDM in the resistance towards cefiderocol has been described for E. cloacae recently (Nurjadi et al. New Delhi metallo-beta-lactamase facilitates the emergence of cefiderocol resistance in Enterobacter cloacae. Antimicrob Agents Chemother. 2021 Dec 6: AAC0201121. doi: 10.1128/AAC.02011-21), this may be included in the discussion as it seems to be an effect of the type of the β -lactamase which may be independent of the bacterial species.

Thanks. We agree with you. Many cefiderocol-resistant strains reported so far harbored NDM. Therefore, the clinical use of cefiderocol in the treatment of NDM producing strains should give sufficient risk prediction of resistance. We also added this research to the discussion section. Please see line 367-371 in the clean version.

9. The limitations sections needs to be expanded. Please be more critical. Was there bias in sampling and is the data representative for China? Do your experiments on *cirA* and *pbp3* really confirm the resistance mechanism to cefiderocol? Etc.

We very much agree with your suggestion. Although our study is a multi-center study, it is not comprehensive enough at the regional and hospital level, and there is still some bias in the data. We have expanded the limitations sections. Please see line 391-396 in the clean version. We have added the *cirA* gene knockout experiment of a clinical NDM-5-producing *E. coli*, and the results proved that the *cirA* gene deletion caused a high cefiderocol resistance. Please see the relevant test results in Table 3.

10. Tables 1 and 2, line 527. MIC 50, MIC 90, this refers to the MIC where 50 or 90% of the isolates were inhibited. Please describe correctly. Why is there a line empty before cefotaxime in table 1 and before cefoperazone/sulbactam in table 2?

Thanks, it has been edited. A line empty before cefotaxime in table 1 and before cefoperazone/sulbactam in table 2 may be a formatting issue; we have fixed it.

11. All tables with MIC values: please mention the unity (mg/L or µg/ml as I suppose).

Thanks. We have unified the MIC units throughout the text.

12. Please expand all figure legends/captions.

a. Figure 1: change title to "Distribution of the most frequent carbapenemases carried...". Legend: green: "no carbapenemase detected by PCR". Please explain the figure to a minimal extent in the caption.

b. Figure 2: Expand the legend to the meaning of the colors, does the color mean the gene is present or absent? The genes need to be named in the legend as well. The scale for the genomic distance seems to be truncated and the unit is missing in the picture. I would further suggest to change the expression "comparison diagram..." to "detected antimicrobial resistance genes".

Thanks. We have revised the legend and description as you suggested. Please see Figure 1 and Figure 2 in the clean version.

Minor comments:

Line 42: „aztreonam-avibactam had..."

Thanks, it has been edited. Please see line 41 in the clean version.

Line 48:" ..., the MIC of the transformants

Thanks, it has been edited. Please see line 47 in the clean version.

Line 50: delete "was"

Corrected.

Line 71-72 Indeed tigecycline is not useful for all types of infection, especially not for blood stream infections. However, intraabdominal infections and especially those affecting the bile duct system can be treated successfully as the concentration in bile is comparably high. Please rephrase the sentence in your manuscript, as it is not completely correct as it is written.

Thanks, it has been edited. Please see lines 81-84 in the clean version.

Line 72: colistin also has a neurotoxicity which is clinically relevant. Please insert this information.

Thanks, it has been edited. Please see lines 84-86 in the clean version.

Line 75: please change to "... it is ineffective against metallo- β -lactamase bearing gram-negative bacilli".

Thanks, it has been edited. Please see line 88 in the clean version.

Line 80: Please cite a work of Marvin J Miller when referring to his work and group. There's a bunch of publications.

Thanks. We have cited the work of Marvin J Miller in the original text. Please see reference (6) in the clean version.

Line 99: I do not agree that it is possible to "eradicate" CRE. It is a promising treatment option. Please rephrase.

Thanks, it has been edited. Please see line 111 in the clean version.

Lines 193 and 204: What is an "appropriate amount"? Please specify.

Thanks, it has been edited. Please see lines 204 and 213 in the clean version.

Line 219: is suppose you mean these isolates had a positive result in the PCR for carbapenemases.

Yes. We have revised the description there to make it easier to read

Line 221: Please change "produced" to detected. And lines 225 and following: change "produced" to harbored or carried.

Thanks, it has been edited.

Line 271: please change to "...contains a stop codon." Please also consider changing the term stopped encoding to a stop coding or any other expression that is commonly used.

Thanks, it has been edited.

Line 274 ff.: Phylogenetic tree is commonly used as expression here, while evolutionary tree is an older- however still correct- expression.

Thanks. Our description is not accurate. We have revised the original text.

Line 279: Figure instead of Figura

Corrected.

Line 298, susceptibility instead of sensitivity

Thanks, it has been edited.

Line 347: I think you mean "An essential issue in the Trojan Horse strategy is if the trojan horse is unable to enter the city"

Thanks, it has been edited. Please see line 386 in the clean version.

Line 360, following "Klein et al." This sentence does not make sense as it is written, please rephrase.

Thanks, it has been edited. Please see line 398-400 in the clean version.

Line 360 and following, it is more common to not give the full name, instead put Klein et al. etc. for the cited authors/group.

Thanks, it has been edited. We modified all places where full names are used.

April 14, 2022

Prof. Hui Wang
Institute of Medical Technology, Peking University Health Science Center
Xueyuan Road, No.38 Beijing 100191, China
Beijing 100191
China

Re: Spectrum02670-21R1 (Occurrence of high levels cefiderocol resistance in carbapenem-resistant Escherichia coli before its approval in China: A report from China CRE-Network)

Dear Prof. Hui Wang:

Your manuscript has been accepted, and I am forwarding it to the ASM Journals Department for publication. You will be notified when your proofs are ready to be viewed.

Sincerely,

Daria Van Tyne
Editor, Microbiology Spectrum

Journals Department
Supplemental Dataset: Accept